# Sparse and Local Networks for Hypergraph Reasoning

**Guangxuan Xiao**
MIT

**Leslie Pack Kaelbling**
MIT

**Jiajun Wu**
Stanford University

**Jiayuan Mao**
MIT

## Abstract

Reasoning about the relationships between entities from input facts (e.g., whether Ari is a *grandparent* of Charlie) generally requires explicit consideration of other entities that are not mentioned in the query (e.g., the *parents* of Charlie). In this paper, we present an approach for learning to solve problems of this kind in large, real-world domains, using *sparse and local hypergraph neural networks* (SpaLoc). SpaLoc is motivated by two observations from traditional logic-based reasoning: relational inferences usually apply locally (i.e., involve only a small number of individuals), and relations are usually sparse (i.e., only hold for a small percentage of tuples in a domain). We exploit these properties to make learning and inference efficient in very large domains by (1) using a sparse tensor representation for hypergraph neural networks, (2) applying a sparsification loss during training to encourage sparse representations, and (3) subsampling based on a novel information sufficiency–based sampling process during training. SpaLoc achieves state-of-the-art performance on several real-world, large-scale knowledge graph reasoning benchmarks, and is the first framework for applying hypergraph neural networks on real-world knowledge graphs with more than 10k nodes.

## 1 Introduction

Performing graph reasoning in large domains, such as predicting the relationship between two entities based on facts given as input, is an important practical problem that arises in reasoning about many domains, including molecular modeling, knowledge networks, and collections of objects in the physical world [Schlichtkrull et al., 2018b, Veličković et al., 2020, Battaglia et al., 2016]. This paper focuses on an inductive learning-based approach to making predictions in hypergraph reasoning problems. Consider the problem of learning to predict the *grandparent* relationship. Given a dataset of labeled family relationship graphs, we aim to build machine-learning algorithms that learn to predict a specific relationship (e.g., *grandparent*) based on other input relationships, such as *father* and *mother*. A crucial feature of such reasoning tasks is that: in order to predict the relationship between two entities (e.g., whether Ari is a *grandparent* of Charlie), we need to jointly consider other entities (e.g., the *father* and *mother* of Charlie).

A natural approach to this problem is to use *hypergraph neural networks* to represent and reason about higher-order relations: in a hypergraph, a *hyper-edge* may connect more than two nodes. As an example, Neural Logic Machines [NLM; Dong et al., 2019] present a method for solving graph reasoning tasks by maintaining hyperedge representations for all tuples consisting of up to $B$ entities, where $B$ is a hyperparameter. Thus, they can infer more complex finitely-quantified logical relations than standard graph neural networks that only consider binary relationships between entities [Morris et al., 2019b, Barceló et al., 2020]. However, there are two disadvantages of such a dense hypergraph representation. First, the training and inference require considering all entities in a domain simultaneously, such as all of the $N$ people in a family relationship database. Second, they scale exponentially with respect to the number of entities considered in a single type of relationship: inferring the *grandparent* relationship between all pairs of entities requires $O(N^3)$ time and space complexity. In practice, for large graphs, these limitations make the training and inference intractable and hinder the application of methods such as NLMs in large-scale real-world domains.

---

Correspondence to: Guangxuan Xiao and Jiayuan Mao: `{xgx,jiayuanm}@mit.edu`.

G. Xiao et al., Sparse and Local Networks for Hypergraph Reasoning. *Proceedings of the First Learning on Graphs Conference (LoG 2022)*, PMLR 198, Virtual Event, December 9–12, 2022.

To address these two challenges, we draw two inspirations from traditional logic-based reasoning: logical rules (e.g., my parent's parent is my grandparent) usually apply *locally* (e.g., only three people are involved in a grandparent rule), and *sparsely* (e.g., the grandparent relationship is sparse across all pairs of people in the world). Thus, during training and inference, we don't need to keep track of the representation of *all* hyperedges but only the hyperedges that are related to our prediction tasks.

Inspired by these observations, we develop the Sparse and Local Hypergraph Neural Network (SpaLoc) for learning sparse relational representations from data in large domains. First, we present a *sparse* tensor-based representation for encoding hyperedge relationships among entities and extend hypergraph neural networks to this representation. Instead of storing a dense representation for all hyperedges, it only keeps track of edges related to the prediction task, which exploits the inherent sparsity of hypergraph reasoning. Second, since we do not know the underlying sparsity structures *a priori*, we propose a training paradigm to recover the underlying *sparse* relational structure among objects by regularizing the graph sparsity. During training, the graph sparsity measurement is used as a soft constraint, while during inference, SpaLoc uses it to explicitly prune out irrelevant edges to accelerate the inference. Third, during both training and inference, SpaLoc focuses on a *local* induced subgraph of the input graph, instead of considering all entities and their relations. This is achieved by a novel sub-graph sampling technique motivated by *information sufficiency* (IS). IS quantifies the amount of information in a sub-graph for predictions about a specific hyperedge. Since the information in a sub-sampled graph may be insufficient for predicting the relationship between a pair of entities, we also propose to use the information sufficiency measure to adjust training labels.

We study the learning and generalization properties of SpaLoc on a domain of relational reasoning in family-tree datasets and evaluate its performance on real-world knowledge-graph reasoning benchmarks. First, we show that, with our sparsity regularization, the computational complexity for inference can be reduced to the same order as the human expert-developed inference method, which significantly outperforms the baseline models. Second, we show that training via sub-graph sampling and label adjustment enables us to learn relational representations in real-world knowledge graphs with more than 10K nodes, whereas other hypergraph neural networks can barely be applied to graphs with more than 100 nodes. SpaLoc achieves state-of-the-art performance on several real-world knowledge graph reasoning benchmarks, surpassing several existing binary-edge-based graph neural networks. Finally, we show the generality of SpaLoc by applying it to different hypergraph neural networks.

## 2   Related Work

**(Hyper-)Graph representation learning**. (Hyper-)Graph representation learning methods, including message passing neural networks [Shervashidze et al., 2011, Kipf and Welling, 2017, Velickovic et al., 2018, Hamilton et al., 2017] and embedding-based methods [Bordes et al., 2013a, Yang et al., 2015, Toutanova et al., 2015, Dettmers et al., 2018], have been widely used for knowledge discovery. Since these methods treat relations (edges) as fixed indices for node feature propagation, their computational complexity is usually small (e.g., $O(NE)$), and they can be applied to large datasets. However, the fixed relation representation and low complexity restrict the expressive power of these methods [Xu et al., 2019, 2020, Luo et al., 2021], preventing them from solving general complex problems such as inducing rules that involve more than three entities. Moreover, some widely used methods, such as knowledge embeddings [Bordes et al., 2013b, Ren et al., 2020], are inherently transductive and cannot learn lifted rules that generalize to unseen domains. By contrast, the learned rules from SpaLoc are inductive and can be applied to completely novel domains with an entire collection of new entities, as long as the underlying patterns of relational inference remain the same.

**Inductive rule learning**. In addition to graph learning frameworks, many previous approaches have studied how to learn generalized rules from data, i.e., inductive logic programming (ILP) [Muggleton, 1991, Friedman et al., 1999], with recent work integrating neural networks into ILP systems to combat noisy and ambiguous inputs [Dong et al., 2019, Evans and Grefenstette, 2018, Sukhbaatar et al., 2015]. However, due to the large search space of target rules, the computational and memory complexities of these models are too high to scale up to many large real-world domains. SpaLoc addresses this scalability problem by leveraging the sparsity and locality of real-world rules and thus can induce knowledge with local computations.

**Efficient training and inference methods**. There is a rich literature on efficient training and inference of neural networks. Two directions that are relevant to us are model sparsification and sampling

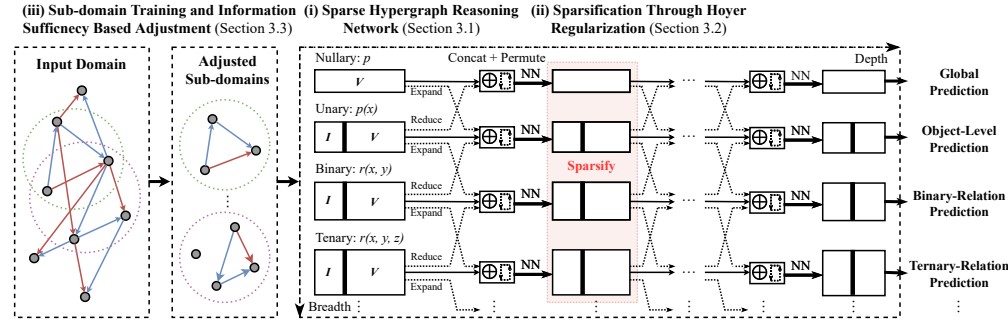

**Figure 1:** The overall training pipeline of SpaLoc, including sub-graph sampling with label adjustment (Sec. 3.3), sparse hypergraph neural networks (Sec. 3.1), and sparsity regularizations (Sec. 3.2). $I$ and $V$ denote the index tensor and value tensor, respectively.

training. Han et al. [2016] proposed to prune and compress the weights of neural networks for efficiency, and Yang et al. [2020] adopted Hoyer-Square regularization to sparsify models. SpaLoc extends this sparsification idea by adding regularization at intermediate sparse tensor groundings to encourage sparse induction. Chiang et al. [2019] and Zeng et al. [2020] proposed to sample sub-graphs for GNN training and Teru et al. [2020] proposed to construct sub-graphs for link prediction. Cheng et al. [2021] uses pre-defined Horn rules to sample subgraphs to train knowledge graph embeddings. SpaLoc generalizes these sampling methods to hypergraphs and proposes the information sufficiency-based adjustment method to remedy the information loss introduced by sub-sampling.

## 3 SpaLoc Hypergraph Neural Networks

This section develops a training and inference framework for hypergraph neural networks. As illustrated in Fig. 1, we make hypergraph networks practical for large domains by using sparse tensors (Sec. 3.1). To encourage models to discover sparse interconnections, we add sparsity regularization to intermediate tensors (Sec. 3.2). We exploit the locality of the task by sampling subgraphs and compensate for information loss due to sampling through a novel label adjustment process (Sec. 3.3).

The fundamental structures used for both training and inference are hypergraphs $\mathcal{H} = (\mathcal{V}, \mathcal{E})$, where $\mathcal{V}$ is a set of vertices and $\mathcal{E}$ is a set of hyperedges. Each hyperedge $e = (x_1, x_2, \cdots, x_r)$ is an ordered tuple of $r$ elements ($r$ is called the arity of the edge), where $x_i \in \mathcal{V}$. We use $f : \mathcal{E} \to \mathcal{S}$ to denote a *hyperedge representation function*, which maps hyperedge $e$ to a feature in $\mathcal{S}$. Domains $\mathcal{S}$ can be of various forms, including discrete labels, numbers, and vectors. For simplicity, we describe features associated with arity-1 edges as "node features" and features associated with the whole graph as "nullary" or "global" features.

A graph-reasoning task can be formulated as follows: given $\mathcal{H}$ and the input hyperedge representation functions $f$ associated with all hyperedges in $\mathcal{E}$, such as node types and pairwise relationships (e.g., *parent*), our goal is to infer a target representation function $f'$ for one or more hyperedges, i.e. $f'(e)$ for some $e \in \mathcal{E}$, such as predicting a new relationship (e.g., *grandparent(Kim,Skye)*). We consider two problem settings in this paper. The first one is to predict a target relation over all edges in the graph. The second one is to predict the relation on one single edge.

### 3.1 Sparse Hypergraph Neural Networks

SpaLoc is a general formulation that can be applied to a range of hypergraph reasoning frameworks. We will primarily develop our method based on the Neural Logic Machine [NLM; Dong et al., 2019], a state-of-the-art inductive hypergraph neural network. We choose an NLM as the backbone network in SpaLoc because its tensor representation naturally generalizes to sparse cases. In Sec. 4.1, we also integrate SpaLoc with other hypergraph neural networks like k-GNNs [Morris et al., 2019a].

In SpaLoc, hypergraph features such as node features and edge features are represented as sparse tensors. For example, as shown in Fig. 1, at the input level, the parental relationship can be represented as a list of indices and values. In this case, each index $(x, y)$ is an ordered pair of integers, and the corresponding value is 1 if node $x$ is a parent of node $y$. To leverage the sparsity in relations, we treat

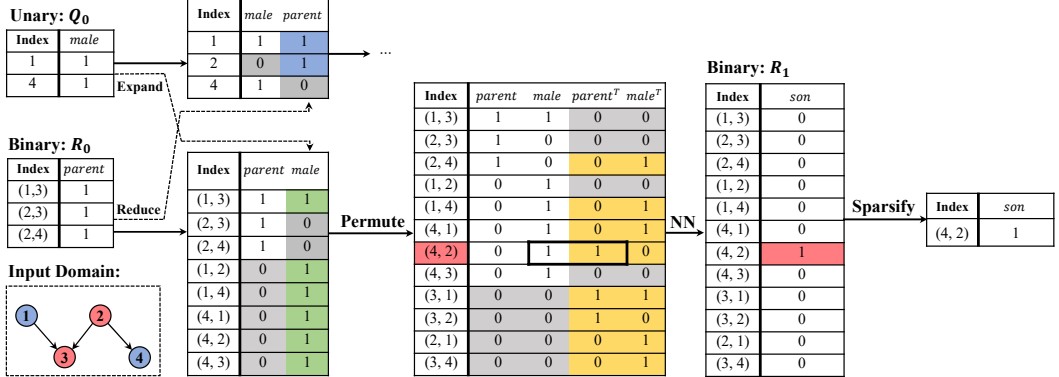

**Figure 2:** A running example of a single layer SpaLoc: inferring the binary relationship of *son(x, y) := male(x) ∧ parent(y, x)* from the attribute *male* and the binary relationship *parent*. The model first expands the unary tensor (containing the *male* information) into a binary relation, indicating whether the first entity in the pair is a male. Then, the permutation operation fuses the information for $(x, y)$ and $(y, x)$. For each pair (x, y), we now have four predicates: whether x is a parent of y, whether y is a parent of x, whether x is a male, and whether y is a male. Finally, a neural network predicts the target relationship *son* for each pair $(x, y)$. Blue entries denote values that are reduced from high-arity tensors. Green entries are expanded from low-arity tensors. Yellow entries are created by the "permutation" operation. Gray entries are zero paddings.

values for indices not in the list as 0. This convention also extends to vector representations of nodes and hyperedges. In general, vector representations $f(x_1, x_2, \cdots, x_r)$ associated with all hyperedges of arity $r$ are represented as coordinate-list (COO) format sparse tensors [Fey and Lenssen, 2019]. That is, each tensor is represented as two tensors $\mathcal{F} = (\mathbf{I}, \mathbf{V})$, each with $M$ entries. The first tensor $\mathbf{I}$ is an *index* tensor, of shape $M \times r$, in which each row denotes a tuple $(x_1, x_2, \cdots, x_r)$. The second tensor $\mathbf{V}$ is a *value* tensor, of shape $M \times D$, where $D$ is the length of $f(x_1, x_2, \cdots, x_r)$. Each row $\mathbf{V}[i]$ denotes the vector representation associated with the tuple $\mathbf{I}[i]$. For all tuples that are not recorded in $\mathbf{I}$, their representations are treated as all-zero vectors.

Based on the sparse feature representations, a sparse hypergraph neural network is composed of multiple *relational reasoning layers* (RRLs) that operate on hyperedge representations. Fig. 1 shows the detailed computation graph of an SpaLoc model with ternary relations. The input to first RRL is the input information (e.g., demographic information and parental relationships in a person database). Each RRL computes a set of new hyperedge features as inputs to the next layer. The last layer output will be the final prediction of the task (e.g., the "son" relationship). During training time, we will supervise the network with ground-truth labels for final predictions.

Next, we describe the computation of individual RRLs. The descriptions will be brief and focus on differences from the original NLM layers. The input to and output of each RRL are both $R + 1$ sparse tensors of different arities, where $R$ is the maximum arity of the network. Let $\mathcal{F}^{(i-1,r)}$ denote the input of arity $r$ of layer $i$, the output of this layer $\mathcal{F}^{(i,r)}$ is computed as the following:

$$\mathcal{F}^{(i,r)} = \text{NN}^{(i,r)} \left( \text{PERMUTE} \left( \text{CONCAT} \left( \mathcal{F}^{(i-1,r)}, \text{EXPAND} \left( \mathcal{F}^{(i-1,r-1)} \right), \text{REDUCE} \left( \mathcal{F}^{(i-1,r+1)} \right) \right) \right) \right)$$

In a nutshell, the EXPAND operation propagates representations from lower-arity tensors to a higher-arity form (e.g., from each node to the edges connected to it). The REDUCE operation aggregates higher-arity representations into a lower-arity form (e.g., aggregating the information from all edges connected to a node into that node). The PERMUTE operation fuses the representations of hyperedges that share the same set of entities but in different orders, such as $(A, B)$ and $(B, A)$. Finally, NN is a linear layer with nonlinear activation that computes the representation for the next layer. Fig. 2 gives a concrete running example of a single RRL.

Formally, the EXPAND operation takes a sparse tensor $\mathcal{F}$ of arity $r$ and creates a new sparse tensor $\mathcal{F}'$ with arity $r + 1$. This is implemented by duplicating each entry $f(x_1, \cdots, x_r)$ in $\mathcal{F}$ by $N$ times, creating the $N$ new vector representations for $(x_1, \cdots, x_r, o_i)$ for all $i \in \{1, 2, \cdots, N\}$, where $N$ is the number of nodes in the hypergraph.

The REDUCE operation takes a sparse tensor $\mathcal{F} = (\mathbf{I}, \mathbf{V})$ of arity $r$ and creates a new sparse tensor $\mathcal{F}'$ with arity $r - 1$: it aggregates all information associated with all $r$-tuples: $(x_1, x_2, \cdots, x_{r-1}, ?)$ with the same $r - 1$ prefix. In SpaLoc, the aggregation function is chosen to be *max*. Thus,

$$f'(x_1, \cdots, x_{r-1}) = \max_{z:(x_1, \cdots, x_{r-1}, z) \in \mathbf{I}} f(x_1, \cdots, x_{r-1}, z).$$

The CONCAT operation concatenates the input hyperedge representations along the channel dimension (i.e., the dimension corresponding to different relational features). Specifically, it first adds missing entries with all-zero values to the input hyperedge representations so that they have exactly the same set of indices $\mathbf{I}$. It then concatenates the $\mathbf{V}$'s of inputs along the channel dimension.

The PERMUTE operation takes a sparse tensor $\mathcal{F}$ of arity $r$ and creates a new sparse tensor $\mathcal{F}'$ of the same arity. However, the length of the vector representation will grow from $D$ to $D' = r! \times D$. It fuses the representation of hyperedges that share the same set of entities. Mathematically,

$$f'(x_1, \cdots, x_r) = \underset{(x'_1, \cdots, x'_r) \text{ is a permutation of } (x_1, \cdots, x_r)}{\text{CONCAT}} [f(x'_1, \cdots, x'_r)].$$

If a permutation of $(x_1, \cdots, x_r)$ does not exist in $\mathcal{F}$, it will be treated as an all-zero vector. Thus, the number of entries $M$ may increase or remain unchanged.

Finally, the $i$-th sparse relational reasoning layer has $R + 1$ linear layers $L^{(i,0)}, L^{(i,1)}, \cdots, L^{(i,R)}$ with nonlinear activations (e.g., ReLU) as submodules with arities 0 through $R$. For each arity $r$, we will concatenate the feature tensors expanded from arity $r - 1$, those reduced from arity $r + 1$, and the output from the previous layer, apply a permutation, and apply $L^{(i,r)}$ on the derived tensor.

To make the intermediate features $\mathcal{F}^{(i,r)}$ sparse, SpaLoc uses a gating mechanism. In SpaLoc, for each linear layer $L^{(i,r)}$, we add a linear gating layer, $L_g^{(i,r)}$, which has sigmoid activation and outputs a scalar value in range $[0, 1]$ that can be interpreted as the importance score for each hyperedge. During training, we modulate the output of $L^{(i,r)}$ with this importance value. Specifically, the output of layer $i$ arity $r$ is $\mathcal{F}^{(i,r)} = L^{(i,r)}(\mathcal{F}) \odot L_g^{(i,r)}(\mathcal{F})$, where $\mathcal{F}$ is the input sparse tensor, and $\odot$ is the element-wise multiplication operation. Note that we are using the same gate value to modulate each channel dimension of $L^{(i,r)}(\mathcal{F})$. During inference, we can prune out edges with small importance scores $L_g^{(i,r)} < \epsilon$, where $\epsilon$ is a scalar hyperparameter. We use $\epsilon = 0.05$ in our experiments.

We have described the computation of a sparsified Neural Logic Machine. However, we do not know a *priori* the sparse structures of intermediate layer outputs at training time, nor at inference time before we actually compute the output. Thus, we have to start from the assumption of a fully-connected dense graph. In the following sections, we will show how to impose regularization to encourage learning sparse features. Furthermore, we will present a subsampling technique to learn efficiently from large input graphs.

**Remark**. Even when the inputs have only unary and binary relations, allowing intermediate tensor representations of higher arity to be associated with hyperedges increases the expressiveness of NLMs [Dong et al., 2019], and Luo et al. [2021] proves that NLMs with max arity $k + 1$ are as expressive as $k$-GNN hypergraph models (note that the regular GNN is 1-GNN). An intuitive example is that, in order to determine the *grandparent* relationship, we need to consider all 3-tuples of entities, even though the input relations are only binary. Despite their expressiveness, hyperedge-based NLMs cannot be directly applied to large-scale graphs. For a graph with more than 10,000 nodes, such as Freebase [Bollacker et al., 2008], it is almost impossible to store vector representations for all of the $N^3$ tuples of arity 3. Our key observation to improve the efficiency of NLMs is that relational rules are usually applied *sparsely* (Sec. 3.2) and *locally* (Sec. 3.3).

## 3.2 Sparsification through Hoyer Regularization

We use a regularization loss to encourage hyperedge sparsity, which is based on the Hoyer sparsity measure (2004). Let $x$ (in our case, the edge gate $g(x_1, \cdots, x_r)$) be a vector of length $n$. Then

$$Hoyer(x) = \frac{(\sum_i^n |x_i|)/\sqrt{\sum_i^n x_i^2} - 1}{\sqrt{n} - 1}.$$

The Hoyer measure takes values from 0 to 1. The larger the Hoyer measure of a tensor, the denser the tensor is. In order to assign weights to different tensors based on their size, we use the Hoyer-Square measure [Yang et al., 2020],

$$H_S(x) = \frac{(\sum_i^n |x_i|)^2}{\sum_i^n x_i^2},$$

which ranges from 1 (sparsest) to $n$ (densest). Intuitively, the Hoyer-Square measure is more suitable than $L_1$ or $L_2$ regularizers for graph sparsification since it encourages large values to be close to 1 and others to be zero, i.e., extremity. It has been widely used in sparse neural network training and has shown better performance than other sparse measures [Hurley and Rickard, 2009]. We empirically compare $H_S$ with other sparsity measures in Appendix D.

The overall training objective of SpaLoc is the task objective plus the sparsification loss, $\mathcal{L} = \mathcal{L}_{task} + \lambda\mathcal{L}_{density}$, where $\mathcal{L}_{density}$ is the sum of the $H_S$, divided by the sum of the sizes of these tensors.

## 3.3 Subgraph Training

Regularization enables us to learn a sparse model that will be efficient at *inference* time, but does not address the problem of *training* on large graphs. We describe a novel strategy that substantially reduces training complexity. It is based on the observation that an inferred relation among a set of entities generally depends only on a small set of other entities that are "related to" the target entities in the hypergraph, in the sense that they are connected via short paths of relevant relations.

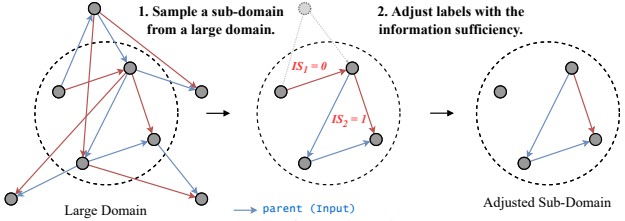

**Figure 3:** Subgraph training contains two steps. First, we sample a subset of nodes from the whole graph. Next, we adjust labels for edges in the sub-sampled graph. $IS_1 = 0$ because no paths connecting two nodes are sampled, while $IS_2 = 1$ because all paths connecting two nodes are sampled.

Specifically, we employ a sub-graph sampling and label adjustment procedure. Here, we first present a measure to quantify the sufficiency of information in a sub-sampled graph for determining the relationship between two entities, namely, *information sufficiency*. Next, we present a sub-graph sampling procedure designed to maximize the information sufficiency for training. We further show that sub-graph sampling can also be employed at inference time. Finally, since information loss is inevitable during sampling, we further propose a training label adjustment process based on the information sufficiency.

**Information sufficiency** Let $\mathcal{H}_S = (\mathcal{V}_S, \mathcal{E}_S)$ be a sub-hypergraph of hypergraph $\mathcal{H} = (\mathcal{V}, \mathcal{E})$, and $e^* = (y_1, \ldots, y_r)$ be a target hyperedge in $\mathcal{H}_S$, where $y_1, \cdots, y_r \in \mathcal{V}_S \subset \mathcal{V}$. Intuitively, in order to determine the label for this hyperedge, we need to consider all "paths" that connect the nodes $\{y_1, \ldots, y_r\}$. More formally, we say a sequence of $K$ hyperedges $(e_1, \ldots, e_K)$, represented as

$$\underbrace{(x_1^1, \cdots, x_{r_1}^1)}_{e_1}, \underbrace{(x_1^2, \cdots, x_{r_2}^2)}_{e_2}, \cdots, \underbrace{(x_1^K, \cdots, x_{r_k}^K)}_{e_K},$$

is a *hyperpath* for nodes $\{y_1, \cdots y_r\}$ if and only if $\{y_1, \cdots y_r\} \subset \bigcup_{j=1}^K e_j$ and $e_j \cap e_{j+1} \neq \emptyset$ for all $j$. In a graph with only binary edges, this is equivalent to the existence of a path from one node $y_1$ to another node $y_2$. We define the *information sufficiency* measure for a hyperedge $e^*$ in subgraph $\mathcal{H}_S$ as ($\frac{0}{0}$ is defined as 1.)

$$IS\left((y_1, \cdots, y_r) \mid \mathcal{H}_S, \mathcal{H}\right) := \frac{\#\text{Paths connecting } (y_1, \cdots, y_r) \text{ in } \mathcal{H}_S}{\#\text{Paths connecting } (y_1, \cdots, y_r) \text{ in } \mathcal{H}}.$$

In practice, we approximate *IS* by only counting the number of paths whose length is less than a task-dependent threshold $\tau$ for efficiency. The number of paths in a large graph can be pre-computed and cached before training, and the overhead of counting paths in a sampled graph is small, so this computation does not add much overhead to training and inference. When input graphs have maximum arity 2, paths can be counted efficiently by taking powers of the graph adjacency matrix.

**Subgraph sampling** During training, each data point is a tuple $(\mathcal{H}, f, f')$ where $\mathcal{H}$ is the input graph, $f$ is the input representation, and $f'$ is the desired output labels. We sample a subgraph $\mathcal{H}' \subset \mathcal{H}$, and train models to predict the value of $f'$ on $\mathcal{H}'$ given $f$. For example, we train models to predict the *grandparent* relationship between all pairs of entities in $\mathcal{H}'$ based on the *parent* relationship

between entities in $\mathcal{H}'$. Thus, our goal is to find a subgraph that retains most of the paths connecting nodes in this subgraph. We achieve this using a *neighbor expansion sampler* that uniformly samples a few nodes from $\mathcal{V}$ as the seed nodes. It then samples new nodes connected with one of the nodes in the graph into the sampled graph and runs this "expansion" procedure for multiple iterations to get $\mathcal{V}_S$. Finally, we include all edges that connect nodes in $\mathcal{V}_S$ to form the final subsampled hypergraph.

When the task is to infer the relations between a single pair of entities $f'(y_1, y_2)$ given the input representation $f$, a similar sub-sampling idea can also be used at inference time to further speed it up. Specifically, we use a *path sampler*, which samples paths connecting $y_1$ and $y_2$ and induces a subgraph from these paths. We provide ablation studies on different sampling strategies in Sec. 4.1. The implementation details of our information sufficiency and samplers are in Appendices C and G.

**Training label adjustment with IS**   Due to the information loss caused by graph subsampling, the information contained in the subgraph may not be sufficient to make predictions about a target relationship. For example, in a family relationship graph, removing a subset of nodes may cause the system to be unable to conclude whether a specific person $x$ has a sibling.

Thus, we propose to adjust the model training by assigning each example $f'(y_1, \cdots, y_r)$ with a soft label, as illustrated in Fig. 3. Consider a binary classification task $f'$. That is, function $f'$ is a mapping from a hyperedge tuple of arity $r$ to $\{0, 1\}$. Denote the model prediction as $\hat{f}'$. Typically, we train the SpaLoc model with a binary cross-entropy loss between $\hat{f}'$ and the ground truth $f'$. In our subgraph training, we instead compute a binary cross-entropy loss between $\hat{f}'$ and $f'_{\mathcal{H}_S} \odot IS$, where $\mathcal{H}_S$ is the sub-sampled graph. Mathematically,
$$\left( f'_{\mathcal{H}_S} \odot IS \right)(y_1, \cdots, y_r) \triangleq f'_{\mathcal{H}_S}(y_1, \cdots, y_r) \cdot IS\left( (y_1, \cdots, y_r) \mid \mathcal{H}_S, \mathcal{H} \right).$$
We empirically compare IS with other label smoothing methods in Appendix F.

## 4   Experiments

In this section, we compare SpaLoc with other methods in two aspects: accuracy and efficiency on large domains. We first compare SpaLoc with other baseline models on a synthetic family tree reasoning benchmark. Since we know the underlying relational rules of the task and have fine-grained control over training/testing distributions, we use this benchmark for ablation studies about the space and time complexity of our model and two design choices (different sampling techniques and different label adjustment techniques). We further extend the results to several real-world knowledge-graph reasoning benchmarks.

### 4.1   Family Tree Reasoning

We first evaluate SpaLoc on a synthetic family-tree reasoning benchmark for inductive logic programming. The goal is to induce target family relationships or member properties in the test domains based on four input relations: *Son*, *Daughter*, *Father*, and *Mother*. Details are defined in Appendix H.

**Baseline**.   We compare SpaLoc against four baselines.   The first three are Memory Networks [MemNNs; Sukhbaatar et al., 2015], $\partial$ILP [Evans and Grefenstette, 2018], and Neural Logic Machines [NLMs; Dong et al., 2019], which are state-of-the-art models for relational rule learning tasks. For these models, we follow the configuration and setup in Dong et al. [2019]. The fourth baseline is an inductive link prediction method based on graph neural networks, GraIL [Teru et al., 2020]. Since GraIL can only be used for link prediction, we use the full-batch R-GCN [Schlichtkrull et al., 2018b], the backbone network of GraIL, for node property predictions.

**Accuracy & Scalability**.   Table 1 summarizes the result. Overall, SpaLoc achieves near-perfect performance across all prediction tasks, on par with the inductive logic programming-based method $\partial$ILP and the baseline model NLM. This suggests that our sparsity regularizations and sub-graph sampling do not affect model accuracy. Importantly, our SpaLoc framework has drastically increased the scalability of the method: SpaLoc can be trained on graphs with 2000 nodes, which is infeasible for the baseline NLM model due to memory issues.

Another essential comparison is between GraIL and SpaLoc. GraIL is a graph neural network–based approach that only considers relationships between binary pairs of entities. This is sufficient for simple tasks such as *HasFather*, but not for more complex tasks such as *Maternal Great Uncle (MGUncle)*. By contrast, SpaLoc explicitly reasons about hyperedges and solves more complex tasks.

**Table 1:** Results (Per-class Accuracy) on family tree reasoning benchmarks. Models are trained on domains with 20 to 2000 entities, and tested on domains with 100 entities. Minus mark means the model runs out of memory or cannot handle ternary predicates. All experiments are conducted on a single NVIDIA 3090 GPU with 24GB memory. The scores and standard errors of SpaLoc are obtained from experiments with three different random seeds.

| Family Tree | MemNN | | $\partial$ILP | | NLM | | GraIL (R-GCN) | | **SpaLoc (Ours)** | |
|---|---|---|---|---|---|---|---|---|---|---|
| $N_{\text{train}}$ | 20 | 2,000 | 20 | 2,000 | 20 | 2,000 | 20 | 2,000 | 20 | 2,000 |
| `HasFather` | 65.24 | - | 100 | - | 100 | - | 100 | 100 | $100_{\pm 0.00}$ | $100_{\pm 0.00}$ |
| `HasSister` | 66.21 | - | 100 | - | 100 | - | 97.05 | 97.95 | $100_{\pm 0.00}$ | $98.01_{\pm 0.04}$ |
| `Grandparent` | 64.57 | - | 100 | - | 100 | - | 99.95 | 98.08 | $100_{\pm 0.00}$ | $100_{\pm 0.00}$ |
| `Uncle` | 64.82 | - | 100 | - | 100 | - | 97.87 | 96.50 | $100_{\pm 0.00}$ | $100_{\pm 0.00}$ |
| `MGUncle` | 80.93 | - | 100 | - | 100 | - | 54.67 | 71.29 | $100_{\pm 0.00}$ | $100_{\pm 0.00}$ |
| `FamilyOfThree` | - | - | - | - | 100 | - | - | - | $100_{\pm 0.00}$ | $100_{\pm 0.00}$ |
| `ThreeGenerations` | - | - | - | - | 100 | - | - | - | $100_{\pm 0.00}$ | $100_{\pm 0.00}$ |

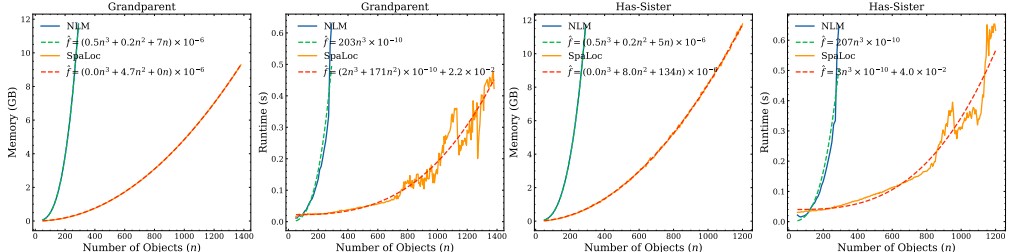

**Figure 4:** The memory usage and the inference time of each sample *vs.* the number of objects in the evaluation domains. SpaLoc reduces the memory complexity from $O(n^3)$ to $O(n^2)$ and achieves significant runtime speedup.

**Runtime & Memory**. We study the time and memory complexity of SpaLoc against NLM on the *HasSister* and *Grandparent* tasks. Results are shown in Fig. 4, where we plot the curve of average memory consumption and inference time as a function of the input graph size. We fit a cubic polynomial equation to the data points to approximate the learned inference complexity of SpaLoc. The experimental results show that our method can empirically reduce the space complexity from the original $O(n^3)$ complexity of NLM to approximately $O(n^2)$. Note that this learned network has the same complexity as the optimal relational rule that can be designed to solve both tasks. The inference time also gets significantly improved.

**Application to other hypergraph neural networks**. SpaLoc is a general framework for scaling up hypergraph neural networks rather than a method that can only be used on NLMs. Here we apply our framework SpaLoc to a new method, k-GNN [Morris et al., 2019a] on the family tree benchmark. Specifically, we use a fully-connected k-hypergraph. The edge embeddings are initialized as a one-hot encoding of the input relationship. Shown in Table 2, we see consistent improvements in terms of inference speed and memory cost for k-GNNs and NLMs.

**Ablation: Subgraph sampling**. We compare our *neighbor expansion sampler* with two other sub-graph samplers, proposed in Zeng et al. [2020]: random node (*Node*) and random walk (*Walk*) samplers. We compare these samplers with two metrics: the final accuracy of the model and the average information sufficiency of all pairs of nodes in the sub-sampled graphs (MIS). Table 3 shows the result on the *Grandparent* task. The *Node* sampler does not leverage locality, so the performance of models and MIS drop as the domain size grows larger. The SpaLocs trained with *Walk* and *Neighbor* samplers perform similarly well in terms of test accuracy. Note that the accuracy results are consistent with the MIS results: comparing the *Node* sampler and others, we see that a higher MIS score leads to higher test accuracy. This supports the effectiveness of our proposed information sufficiency measure.

## 4.2 Real-World Knowledge Graph Reasoning

To further demonstrate the scalability of SpaLoc, we apply it to complete real knowledge graphs. We test SpaLoc on both inductive and transductive relation prediction tasks, following

**Table 2:** Per-class Accuracy, per-sample inference time (ms), and memory usage (MB) when applying SpaLoc on 2-GNNs. Recall that 1-GNN is the standard GNN with only binary edge message passing. Models are tested on domains with 200 entities.

| | Uncle | | | Grandparent | | |
|---|---|---|---|---|---|---|
| | Acc. | Time | Mem. | Acc. | Time | Mem. |
| NLM | 100 | 133.8 | 3,846 | 100 | 135.0 | 3,846 |
| **SpaLoc + NLM** | 100 | **37.2** | **214** | 100 | **23.9** | **181** |
| 2-GNN | 100 | 145.1 | 5,126 | 100 | 145.5 | 5,126 |
| **SpaLoc + 2-GNN** | 100 | **23.7** | **645** | 100 | **19.2** | **519** |

**Table 3:** Comparison of different samplers. The first column shows the size of the sub-sampled graph during training ($N_s$) and the full training graph ($N$). Models are tested on domains with 100 entities.

| $N_s/N$ | Node | | Walk | | Neighbor | |
|---|---|---|---|---|---|---|
| | Acc | MIS | Acc | MIS | Acc | MIS |
| 20 / 50 | 100 | 54.82 | 100 | 85.14 | 100 | 89.78 |
| 20 / 200 | 100 | 33.05 | 100 | 71.51 | 100 | 80.60 |
| 20 / 500 | 58.18 | 27.27 | 100 | 78.22 | 100 | 78.70 |
| 20 / 1,000 | 1.84 | 24.49 | 100 | 77.18 | 100 | 78.38 |
| 20 / 2,000 | 0 | 19.66 | 100 | 79.69 | 100 | 78.53 |

**Table 4:** Results (AUC-PR) on real-world knowledge graph inductive reasoning datasets from GraIL. The scores and standard errors (appendix Table 7) of SpaLoc are obtained from experiments with three different random seeds.

| Model | WN18RR | | | | FB15k-237 | | | | NELL-995 | | | |
|---|---|---|---|---|---|---|---|---|---|---|---|---|
| | v1 | v2 | v3 | v4 | v1 | v2 | v3 | v4 | v1 | v2 | v3 | v4 |
| Neural-LP | 86.02 | 83.78 | 62.90 | 82.06 | 69.64 | 76.55 | 73.95 | 75.74 | 64.66 | 83.61 | 87.58 | 85.69 |
| DRUM | 86.02 | 84.05 | 63.20 | 82.06 | 69.71 | 76.44 | 74.03 | 76.20 | 59.86 | 83.99 | 87.71 | 85.94 |
| RuleN | 90.26 | 89.01 | 76.46 | 85.75 | 75.24 | 88.70 | 91.24 | 91.79 | 84.99 | 88.40 | 87.20 | 80.52 |
| GraIL | 94.32 | 94.18 | 85.80 | 92.72 | 84.69 | 90.57 | 91.68 | 94.46 | 86.05 | 92.62 | 93.34 | 87.50 |
| TACT | 96.15 | 97.95 | 90.58 | 96.15 | 88.73 | 94.20 | 97.10 | 98.30 | 94.87 | 96.58 | 95.70 | 96.12 |
| **SpaLoc** | **98.45** | **99.96** | **96.63** | **99.34** | **99.74** | **99.34** | **99.55** | **99.29** | **100** | **98.56** | **97.19** | **97.34** |

GraIL [Schlichtkrull et al., 2018a]. In this setting, the test-task time is to infer the relationship on a given edge, so test-time graph subsampling is used. We evaluate the models with a classification metric, the area under the precision-recall curve (AUC-PR).

In the inductive setting, the training and evaluation graphs are disjoint sub-graphs extracted from WN18RR [Dettmers et al., 2018], FB15k-237 [Toutanova et al., 2015], and NELL-995 [Xiong et al., 2017]. For each knowledge graph, there are four versions with increasing sizes. In the transductive setting, we use the standard WN18RR, FB15k-237, and NELL-995 benchmarks. For WN18RR and FB15k-237, we use the original splits; for NELL-995, we use the split provided in GraIL. We also include the Hit@10 metric used by knowledge graph embedding methods. Following the setting of GraIL, we rank each test triplet among 50 randomly sampled negative triplets.

**Baseline**. We compare SpaLoc with several state-of-the-art models, including Neural LP [Yang et al., 2017], DRUM [Sadeghian et al., 2019], RuleN [Meilicke et al., 2018], GraIL, and TACT [Chen et al., 2021]. For transductive learning tasks, we compare SpaLoc with four representative knowledge graph embedding methods: TransE [Bordes et al., 2013a], DistMult [Yang et al., 2015], ComplEx [Trouillon et al., 2017], and RotatE [Sun et al., 2019].

**Results**. Table 4 and Table 5 show the inductive and transductive relation prediction results respectively. In the inductive setting, SpaLoc significantly outperforms all baselines on all datasets. This demonstrates the scalability of SpaLoc on large-scale real-world data. SpaLoc is the only model that explicitly uses hyperedge representations, while none of the existing hypergraph neural networks are directly applicable to such large graphs due to memory and time complexities. In the transductive setting, SpaLoc outperforms all knowledge embedding (KE) methods and GraIL on WN18RR and NELL-995. SpaLoc also has comparable performance with KE methods on the FB15K-237 datasets, outperforming GraIL by a large margin. Comparing SpaLoc with node embedding–based methods (TransE) and GNN-based methods (GraIL) that only consider binary edges, we see that our hyperedge-based model enables better relation prediction that requires reasoning about other entities. The necessity of hyperedges is further supported by Appendix E, where we show that setting the maximum arity of SpaLoc to 2 (i.e., removing hyperedges) significantly degrades the performance.

Notably, in contrast to other methods for the transductive setting that store entity embeddings for all knowledge graph nodes, SpaLoc directly uses the inductive learning setting. That is, SpaLoc does

**Table 5:** Results of transductive link prediction on real-world knowledge graphs. We also include Hit@10 as an additional metric following knowledge graph embedding literature and GraIL [Schlichtkrull et al., 2018a]. The scores and standard errors (appendix Table 8) of SpaLoc are obtained from experiments with three different random seeds.

|  | WN18RR | | NELL-995 | | FB15K-237 | |
|---|---|---|---|---|---|---|
|  | AUC-PR | H@10 | AUC-PR | H@10 | AUC-PR | H@10 |
| TransE | 93.73 | 88.74 | 98.73 | 98.50 | 98.54 | 98.87 |
| DistMult | 93.08 | 85.35 | 97.73 | 95.68 | 97.63 | 98.67 |
| ComplEx | 92.45 | 83.98 | 97.66 | 95.43 | 97.99 | **98.88** |
| RotatE | 93.55 | 88.85 | 98.54 | 98.09 | 98.53 | 98.81 |
| GraIL | 90.91 | 73.12 | 97.79 | 94.54 | 92.06 | 75.87 |
| **SpaLoc** | **96.74** | **99.94** | **99.25** | **98.93** | **99.69** | 96.96 |

not store entity information about each knowledge graph node. SpaLoc performs pretty well in the classification setting and ranking setting when negative candidate sets are small (Table 5). However, we find inductive reasoning methods, including SpaLoc, GraIL [Teru et al., 2020], and TACT [Chen et al., 2021], perform poorly in the traditional knowledge graph ranking setting [Bordes et al., 2013b] (i.e., ranking the positive triplets against negative triplets of all replacement of head/tail) entities). The ranking scores are close to zero when the negative candidates set are large. We attribute this phenomenon to the fact that the inductive models do not store entity information. We leave better adaptations of SpaLoc to transductive learning settings as future work.

## 5 Conclusion

We present SpaLoc, a framework for efficient training and inference of hypergraph neural networks for hypergraph reasoning tasks. SpaLoc leverages sparsity and locality to train and infer efficiently. Through regularizing intermediate representation by a sparsification loss, SpaLoc achieves the same inference complexities on family tree tasks as algorithms designed by human experts. SpaLoc samples sub-graphs for training and inference, calibrates training labels with the information sufficiency measure to alleviate information loss, and therefore generalizes well on large-scale relational reasoning benchmarks.

**Limitations.**. The locality assumption applies to many benchmark datasets, but we admit that it is not a completely general solution. It may lead to problems on datasets where the property of interest may depend on distant nodes, i.e., SpaLoc may not perform well on problems that require long chains of inference (e.g., detecting that A is a 5th cousin of B). Nevertheless, the locality assumption is good enough for many real-world relational inference tasks. Meanwhile, it is hard to directly apply SpaLoc on extremely high-arity hypergraph datasets such as WD50K [Galkin et al., 2020], where the maximum arity is 67, because the permutation operation in SpaLoc has an $O(r!)$ complexity, where r is the arity. We leave the application of SpaLoc on extremely high-arity hypergraphs as future work. Additionally, although we show that SpaLoc can empirically reduce the inference time and memory complexity of hypergraph neural networks (Fig. 4), the theoretical bounds for the learned time and memory complexity of SpaLoc remain unclear. Future works can explore the optimal complexity limits of SpaLoc and the conditions to achieve them.

**Acknowledgement.**. This work is in part supported by ONR MURI N00014-22-1-2740, NSF grant 2214177, AFOSR grant FA9550-22-1-0249, ONR grant N00014-18-1-2847, the MIT Quest for Intelligence, MIT–IBM Watson Lab, the Stanford Institute for Human-Centered Artificial Intelligence (HAI), and Analog, JPMC, and Salesforce. Any opinions, findings, and conclusions or recommendations expressed in this material are those of the authors and do not necessarily reflect the views of our sponsors.

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

# SUPPLEMENTARY MATERIAL

The supplementary material is organized as follows. First, we provide the experimental configurations and hyperparameters in Appendix A. Second, we provide the standard errors of SpaLoc's results in Appendix B. In Appendix C, we provide implementation details of the subgraph samplers we used. Next, we provide the ablation study on sparsification loss and SpaLoc's maximum arity in Appendix D, Appendix E, and Appendix F, respectively. Besides, we elaborate the calculation of information sufficiency in Appendix G. Finally, we define relations in the Family Tree reasoning benchmark in Appendix H.

## A    Experimental configuration

**Table 6:** Hyper-parameters for SpaLoc.

| | Tasks | Depth | Breadth | Hidden Dims | Batch size | Subgraph size | $\tau$ |
|---|---|---|---|---|---|---|---|
| Family Tree | HasFather | 5 | 3 | 8 | 8 | 20 | 1 |
| | HasSister | 5 | 3 | 8 | 8 | 20 | 2 |
| | Grandparent | 5 | 3 | 8 | 8 | 20 | 2 |
| | Uncle | 5 | 3 | 8 | 8 | 20 | 2 |
| | MGUncle | 5 | 3 | 8 | 8 | 20 | 2 |
| | Family-of-three | 5 | 3 | 8 | 8 | 20 | 2 |
| | Three-generations | 5 | 3 | 8 | 8 | 20 | 2 |
| Inductive KG | WN18RR | 6 | 3 | 64 | 128 | 10 | 3 |
| | FB15K-237 | 6 | 3 | 64 | 64 | 20 | 3 |
| | NELL-995 | 6 | 3 | 64 | 128 | 10 | 3 |
| Transductive KG | WN18RR | 6 | 3 | 64 | 64 | 20 | 3 |
| | FB15K-237 | 6 | 3 | 64 | 64 | 20 | 3 |
| | NELL-995 | 6 | 3 | 64 | 64 | 20 | 3 |

We optimize all models with Adam [Kingma and Ba, 2015] and use an initial learning rate of 0.005. All experiments are under the supervised learning setup; we use Softmax-Cross-Entropy as the loss function.

Table A shows hyper-parameters used by SpaLoc. For all MLP inside SpaLoc, we use no hidden layer and the sigmoid activation. Across all experiments in this paper, the maximum arity of intermediate predicates (i.e., the "breadth") is set to 3 as a hyperparameter, which allows SpaLoc to realize all first-order logic (FOL) formulas with at most three variables, such as a "transitive relation rule." We set the specification threshold $\epsilon$ to $0.05$ in our experiments. This value is chosen based on the validation accuracy of our model. In practice, we observe that any values between 0.01 and 0.1 do not significantly impact the performance of our method and the inference complexity. We also set the multiplier of the sparsification loss $\lambda$ to $0.01$ in all SpaLoc's experiments.

## B    SpaLoc's results with standard errors

We show the standard errors of SpaLoc's results in Table 7 and Table 8. The scores and standard errors are obtained from experiments with three different random seeds. We find the standard erros are small, i.e., the performance of SpaLoc is quite stable.

## C    Implementation of subgraph samplers

Both the neighbor expansion and the path sampler sample subgraphs by inducing from selected node sets. To deal with input hypergraphs with any arities, the samplers simplify the input hypergraphs into binary graphs. We define two nodes in the hypergraph are connected if they are covered by a hyperedge. Therefore, the neighbor expansion and path-finding algorithms used by the samplers can be applied to any hypergraphs for finding node sets. After enough nodes are collected, the sampler will induce a sub-hypergraph from the original hypergraph by preserving all of the hyperedges lying in the set.

**Table 7:** SpaLoc's results with standard errors on real-world knowledge graph inductive reasoning datasets from GraIL (Table 4).

| | WN18RR | | | | FB15k-237 | | | | NELL-995 | | |
|---|---|---|---|---|---|---|---|---|---|---|---|
| v1 | v2 | v3 | v4 | v1 | v2 | v3 | v4 | v1 | v2 | v3 | v4 |
| $98.45_{\pm 0.20}$ | $99.96_{\pm 0.04}$ | $96.63_{\pm 0.05}$ | $99.34_{\pm 0.01}$ | $99.74_{\pm 0.02}$ | $99.34_{\pm 0.03}$ | $99.55_{\pm 0.01}$ | $99.29_{\pm 0.08}$ | $100_{\pm 0.00}$ | $98.56_{\pm 0.00}$ | $97.19_{\pm 0.08}$ | $97.34_{\pm 0.06}$ |

**Table 8:** SpaLoc's results with standard errors of transductive link prediction on real-world knowledge graphs (Table 5).

| WN18RR | | NELL-995 | | FB15K-237 | |
|---|---|---|---|---|---|
| AUC-PR | Hit@10 | AUC-PR | Hit@10 | AUC-PR | Hit@10 |
| $96.74_{\pm 0.02}$ | $99.94_{\pm 0.03}$ | $99.25_{\pm 0.04}$ | $98.93_{\pm 0.04}$ | $99.69_{\pm 0.05}$ | $96.96_{\pm 0.04}$ |

**Table 9:** Comparison of different sparsification losses.

| | HasSister | | Grandparent | | Uncle | |
|---|---|---|---|---|---|---|
| | Accuracy | Density | Accuracy | Density | Accurcay | Density |
| $L_1$ | 91.81 | 0.48% | 99.8% | 0.99% | 74.69% | 0.68% |
| $L_2$ | 100 | 0.75% | 100% | 0.61% | 94.46% | 2.44% |
| $H_S$ | 100 | 0.51% | 100% | 0.48% | 100% | 0.87% |

## D  Ablation study on sparsification loss

We compare our Hoyer-Square sparsification loss against the $L_1$ and $L_2$ regularizers on the family tree datasets. In Table 9, we show the performance of SpaLoc trained with different sparsification losses. All models are tested on domains with 100 objects.

"Density" is the percentage of non-zero elements (NNZ) in the intermediate groundings. The lower the density, the better the sparsification and the lower the memory complexity. We can see that, compared with $L_1$ and $L_2$ regularizers, the Hoyer-Square loss yields a higher or comparable sparsity while maintaining nearly perfect accuracy.

## E  Ablation study on SpaLoc's maximum arity

In this section, we compare two different SpaLoc models with different maximum arity, to validate the necessity and effectiveness of hyperedges in inductive reasoning. Shown in Table 10 and Table 11, we compare SpaLoc models with max arity 2 and 3. Note that, even if the input and output relations are binary, adding ternary edges in the intermediate representations significantly improves the result.

## F  Ablation Study on Label Adjustment

In this section, we compare our information sufficiency-based label adjustment method (IS) against two simple baselines: no adjustment ("NC"), and label smoothing (LS). In "LS", we multiply all positive labels with a constant $\alpha = 0.9$.

Results are shown in Table 12. Overall, our method (IS) outperforms both baselines, even when the average information sufficiency of training graphs is very low (e.g., when using the *Node* sampler). Especially on the *HasFather* task, using constant label smoothing only has a close-to-chance accuracy (50%). Combining our IS-based label adjustment with the *Neighbor* sampler yields the best result.

## G  Calculation of the information sufficiency

The crucial part in the computation of the information sufficiency is to count the number of $k$-hop hyperpaths connecting a given set of nodes $\{v_1, \ldots, v_r\}$ in a hypergraph. We use the incidence matrix to calculate this. Firstly, we use a $n \times m$ incidence matrix $B$ to represent the hypergraph

**Table 10:** Comparison (Per-class Accuracy) of SpaLoc with different max arities on family tree reasoning benchmarks.

|  | HasSister | Grandparent | Uncle |
|---|---|---|---|
| Max Arity = 2 | 87.66 | 86.74 | 50.00 |
| Max Arity = 3 | **100** | **100** | **100** |

**Table 11:** Comparison (AUC-PR) of SpaLoc with different max arities on real-world knowledge graph inductive relation prediction task.

|  | WN18RR-v1 | FB15k-237-v1 | NELL-995-v1 |
|---|---|---|---|
| Max Arity = 2 | 97.65 | 90.71 | 94.58 |
| Max Arity = 3 | **98.18** | **99.73** | **100** |

**Table 12:** Comparison (per-class accuracy) for different label calibration methods.

| Sampler | HasFather | | | HasSister | | |
|---|---|---|---|---|---|---|
|  | NC | LS | IS | NC | LS | IS |
| *Node* | 50.00 | 50.00 | 50.00 | 50.00 | 50.00 | **80.72** |
| *Walk* | 50.00 | 52.41 | **100** | 59.90 | 75.13 | **93.16** |
| *Neighbor* | 50.00 | 51.63 | **100** | 75.29 | 78.06 | **98.01** |

$\mathcal{H} = (\mathcal{V}, \mathcal{E})$, where $n = |\mathcal{V}|$ and $m = |\mathcal{E}|$, such that $B_{ij} = 1$ if the vertex $v_i$ and edge $e_j$ are incident and 0 otherwise. Next, $B^{(k)} := (BB^T)^{k-1}B$ is the $k$-hop incidence matrix of the hypergraph, i.e., $B_{ij}^{(k)}$ is the number of $k$-hop paths that the vertex $v_i$ and edge $e_j$ are incident.

For example, when $r = 2$, there are $B_i^{(k)} B_j^T$ $k$-hop paths connecting vertex $v_i$ and $v_j$. When $r = 1$, there are $\sum_j B_{ij}^{(k)}$ $k$-hop paths connecting to vertex $v_i$.

## H   Definition of Relations in Family Tree

The inputs predicates are: $\texttt{Father}(x, y)$, $\texttt{Mother}(x, y)$, $\texttt{Son}(x, y)$, $\texttt{Daughter}(x, y)$.

The target predicates are:

- $\texttt{HasFather}(x) := \exists a \ \texttt{Father}(x, a)$
- $\texttt{HasSister}(x) := \exists a \exists b \ \texttt{Father}(x, a) \wedge \texttt{Daughter}(a, b) \wedge \neg(b = x)$
- $\texttt{Grandparent}(x, y) := \exists a \ \texttt{parent}(x, a) \wedge \texttt{parent}(a, y)$
  $\texttt{parent}(x, y) := \texttt{Father}(x, y) \vee \texttt{Mother}(x, y)$
- $\texttt{Uncle}(x, y) := \exists a \ \texttt{Grandparent}(x, a) \wedge \texttt{Son}(a, y) \wedge \neg\texttt{Father}(x, y)$
- $\texttt{MGUncle}(x, y) := \exists a \exists b \ \texttt{Grandmother}(x, a) \wedge \texttt{Mother}(a, b) \wedge \texttt{Son}(b, y)$
  $\texttt{Grandmother}(x, y) = \exists a \ \texttt{Parent}(x, a) \wedge \texttt{Mother}(a, y)$
- $\texttt{Family-of-three}(x, y, z) = \texttt{Father}(x, y) \wedge \texttt{Mother}(x, z)$
- $\texttt{Three-generations}(x, y, z) = \texttt{Parent}(x, y) \wedge \texttt{Parent}(y, z)$

We follow the dataset generation algorithm presented in Dong et al. [2019]. In detail, we simulate the growth of families to generate examples. For each new family member, we randomly assign gender and a pair of parents (can be none, which means it is the oldest person in the family tree) for it. After generating the family tree, we label the relationships according to the definitions above.

