# OpenReview forum: "Sparse and Local Networks for Hypergraph Reasoning"
_logconference.io/LOG/2022/Conference — LoG 2022 Poster_

### Official Review · Reviewer_6ngv · 2022-10-12

**Overall Score:** 6
**Confidence:** 4

**Review:**

Summary:
The paper introduces a framework for learning and inference in hypergraph large scale domains. A sparse representation of hypergraphs and corresponding neural modules is discussed, together with a sparsifiyng regularization loss. Locality arguments are introduced, which allow to scale by exploiting sub-sampling techniques for the input hypergraph. Experiments show that the proposed method outperforms other inductive hypergraph techniques and transductive knowledge graph embedding techniques.


Strong points:
- The method proposes a sound solution to scale in hypergraph learning and inference tasks
- The paper is clearly written

Weak points:
- The main messageof the paper is not clear (is it the architecture? the scaling strategy?)
- Consequently, related work discussion and competitors are not selected accordingly

For these reasons, I recommend a weak reject. I think that the paper is a sound and meaningful strategy, but it is hard from the current shape of the paper and of the experiments to measure the novelty.


DETAILED FEEDBACKS:

- POSITIONING: The paper is a bit unclear about the main message of the work. Abstract and introduction often mention "rule learning", or "rule that explains", which could give a completely wrong impression of what the paper is eventually about. Also the use of the term "reasoning networks", which, without a clear definition, recalls standard reasoning techniques. Instead, for me the paper is much more focused on scaling hypergraph neural networks. And here comes my first doubt: from the current paper (both related works and experiments) it is hard (if not impossible) to position the proposed method in the literature. Because many competitors do not focus on scaling and are not scaling/sub-sampling strategies, but standard (hyper)graph representation techniques. The experiment that goes more in this direction is the ablation study with multiple sampling strategies. But again it is unclear what is the novelty there, as those techniques are recovered from the literature.

- CONTRASTIVE COMPARISONS: Connected to the previous comment, it seems to me that many of the choises made are very interesting / sound, but not discussed and not compared critically with other solutions in the literature. For example, the "permutation" layer is kind of building a "propositionalized" representation of the tuple (a flat representation, with permutations explicitely encoded in the representation). I think this has to be a core part of this relational representation (which is in a certain sense the opposite of what GNNs tend to do) but it is hard to understand the reasons behind such choice.

- OTHER METHODS USING SAMPLING: Still connected to previous points, the discussion about other sampling methods and strategies is not discussed in enough details. I think this is one of (if not the) main contributions of the paper. Scalability is reached by means of sampling. While not being an expert in the specific area (subsampling), many knowledge graph methods with logical rules (which creates hyperedge features) are strongly related to the method, as they need (often locality-based) heuristics to scale to large graphs. E.g. UniKER: A Unified Framework for Combining Embedding and Definite Horn Rule Reasoning for Knowledge Graph Inference, EMNLP 2021, and some of the references in the paper. In this paper, horn rules (which often take the shape of typed paths) are extracted first and then used to sample training data. Sparsifying heuristics are also used during training (thresholds on confidence levels).

QUESTIONS:
1) What is the main contribution of the paper?
2) Why is it novel and which gap in the literature is it filling?
3) Are there other methods using sampling / local heuristics? How the proposed method compares to them?

SUGGESTION FOR IMPROVEMENTS:
I really believe that the paper makes a nice contribution, but it may require a bit more work.
Two directions seem more important to me:
1) the paper must be positioned better, in such a way to clearly underline the main novelty(-ies).
2) comparisons (both discussions and experiments) must concern closer methods (i.e. other scalable (hyper)-graph techniques).



UPDATE: I thank the authors for their response. I would like to raise my score to weak accept, as the authors tried to place better their work w.r.t. the literature.

---

### Official Review · Reviewer_j32J · 2022-10-21

**Overall Score:** 6
**Confidence:** 3

**Review:**

The paper addresses the problem of predicting relationships between entities in a knowledge graph. Knowledge graphs are sometimes also called heterogeneous information graphs or multi-relational graphs. More specifically, the paper's main focus is on making neural networks for hypergraphs more scalable by inducing and taking advantage of sparsity as well as the locality of the computation which is typically (empirically) sufficient for the reasoning task under consideration. For large graphs, the paper overcomes the problem of intractable inference.

The authors describe their approach using Neural Logic machines as the base model but also describe how their ideas can be applied to other hypergraph representation learning methods such as k-GNNs.At the core of their approach is a second loss term which regularizes the network towards sparse hypergraph representations. Another contribution is a subgraph sampling strategy which improves the efficiency of the method during training, where hypergraphs might not (yet) be sparse.

The experiments are extensive and indicate that SpaLoc might achieve higher evaluation metric scores. A shortcoming, which most link prediction methods have in common, is the missing variance of the results. Since this is (unfortunately) a community standard, I don't want to hold it against the authors' method and setup. A more problematic issue I see with the experiments is that the evaluation appears non-standard and uses randomly chosen negative samples. Compared to the standard evaluation procedure used in the knowledge base completion literature, which always compares all possible substitutions in a triple, this leads to a much simpler task. I've reviewed this paper before, and despite several reviewers mentioning the non-standard evaluation setup, it seems the authors used the same evaluation in the resubmission.

What I would also continue to work on is an improvement of the presentation of the method in section 3. There are several paragraphs and subsections but what I am missing is how these pieces fit together and a coherent story that enables the reader to understand the way these parts build a whole. Some sentences between subsections connecting the various contributions should be added.

---

### Official Review · Reviewer_VSsC · 2022-10-21

**Overall Score:** 6
**Confidence:** 3

**Review:**

The paper describes a graph SpaLoc, a novel graph machine learning method designed to learn from hypergraphs to predict missing links between nodes.

The paper is well organized and reads well.

The research problem is pertinent to the scope of the conference and it is well explained - I appreciate papers that explore learning from hypergraphs.

The authors provide a reasonable overview of the literature landscape, although I would also include NodePiece, [Galkin et al.]).

The novel contribution (the SpaLoc model) is sufficient to meet the acceptance bar. The intuition of leveraging sparsity and locality of rules to induce from data is reasonable, and SpaLoc's architecture implements such ideas in a well-grounded manner.

A note on SpaLoc scalability: sure, the method scales better than other direct competitors, but it is important to underline that all datasets used in the paper are toy datasets (see my question on training time on transductive link prediction datasets).

I have some questions on the evaluation section, perhaps the authors can clarify in the rebuttal:

1) There is a blind spot in comparing with 3rd-party methods, which the authors should address, and that is the comparison against methods for multi-hop query answering on incomplete knowledge graphs (such as Arakelyan et al.'s CQD or QueryToBox). The "Maternal great uncle" example in 303 could be addressed by these methods as well, and a comparison would be interesting. This could be done at least in a qualitative way.

2) On the transductive link prediction experiments, could you clarify better what is the arity hyperparam value you used to train SpaLoc for the results in table5? If you did not remove hyperdeges (360-360), does it mean that the benchmark training sets have been turned into hypergraphs? (all three benchmark datasets are regular knowledge graphs) - I'd just need a clarification on this point, to make sure the protocol you followed was fair to the transductive baselines.

3) Why not adopting the conventional transductive link prediction protocol instead? It is hard to tell if SpaLoc really outperforms the baselines in these conditions (e.g. very low number of synthetic negatives per positive). Besides, reporting other metrics such as MR, MRR, Hits@n, n=1,3 would also help.

4) I could not find hyperparam values and hyperparam ranges of the baselines or SpaLoc. It is important to make sure these values guarantee a fair comparison across the board, as transductive methods are indeed sensitive to these values (see also Ruffinelli et al. ICLR-20)

5) What is the time to train the mode on the transductive link prediction benchmark datasets? Is it worth adopting SpaLoc in practice or KGE baselines still outperform the method in terms of running time?

6) On the inductive link prediction experiment: NodePiece and relational GCN (R-GCN) would also be two good baselines to compare against (Nodepiece being also interesting to compare in terms of memory footprint, as it does not learn representations for each node of the graph either).

All in all, a well-written paper, with a sufficiently good original contribution to an interesting problem.

I have some doubts on the impact of the method (see my questions above), but I am happy to keep my positive score if the rebuttal and discussion phase address the concerns I have listed above.

---

### Official Review · Reviewer_RBbk · 2022-10-22

**Overall Score:** 5
**Confidence:** 5

**Review:**

This paper presents an approach to reasoning about the relations in a knowledge hypergraph using sparse and local hypergraph reasoning. Their main motivation is to make learning and inference efficient in very large domains by utilizing sparse tensor representation for hypergraph neural networks, applying a sparsification loss during training, and subsampling.

Main concerns

* The paper is not well written. While it motivated the problem well, the intuition behind design choices is not clear and needs to be explained in a better way.
* The experiment section needs to be improved. I highly recommend to add more experiments and baselines. Many related work is missing from this paper. For example [1], [2]
[1] Bahare Fatemi, Perouz Taslakian, David Vazquez, and David Poole. 2021. Knowledge hypergraphs: prediction beyond binary relations. In Proceedings of the Twenty-Ninth International Joint Conference on Artificial Intelligence (IJCAI'20).
[2] irui Chen, Xin Wang, Chenxu Wang, and Jianxin Li. 2022. Explainable Link Prediction in Knowledge Hypergraphs. In Proceedings of the 31st ACM International Conference on Information & Knowledge Management (CIKM '22).

* Since one of the contributions of this paper is reducing time and memory, such claims need to be proved and shown by analyzing the algorithms. The experimental numbers are not enough to make such claims.
* Details of the datasets used in the paper should be included. Furthermore, since the paper claims it is targeting very large hypergraphs, such datasets should be added to the experimental results.

---

### Meta-Review · Area_Chair_hsmU · 2022-11-18

**Confidence:** 4
**Recommendation:** Accept

**Meta Review:**

The majority of the reviewers were inclined towards acceptance.
Raised concerns involved the paper's readability (e.g., it's hard to understand the intuition behind some of the design choices), the choice of the experimental settings and baselines, experiments on the computational efficiency of the methods, and the datasets, to which the authors answered in great detail.

---

### Decision · Program_Chairs · 2022-11-22

**Decision:**

Accept (Poster)

**Comment:**

In light of the discussion and the rebuttal, we find that this paper can be accepted. We strongly encourage the authors to address the clarity and accessibility suggestions raised by the reviewers in their revision.